# Simultaneous Influenza Vaccination and Hymenoptera Venom Immunotherapy Is Safe

**DOI:** 10.3390/vaccines9040344

**Published:** 2021-04-02

**Authors:** Ewa Czerwińska, Marita Nittner-Marszalska, Robert Pawłowicz, Leszek Szenborn

**Affiliations:** 1Department of Paediatrics and Infectious Diseases, Wroclaw Medical University, ul. Tytusa Chałubińskiego 2-2a, 50-368 Wroclaw, Poland; leszek.szenborn@umed.wroc.pl; 2Department of Internal Medicine, Pneumology and Allergology, Wroclaw Medical University, ul. Marii Skłodowskiej-Curie 66, 50-369 Wroclaw, Poland; marmarsz@gmail.com (M.N.-M.); robert.pa@wp.pl (R.P.)

**Keywords:** allergen immunotherapy, Hymenoptera venom allergy, influenza vaccination, simultaneous administration

## Abstract

Allergen immunotherapy (AIT) is a standard treatment for venom allergy. Our purpose was to determine if the administration of both allergen and protective vaccines during one visit is safe and if such a procedure does not deteriorate the tolerance of both vaccines. As current guidelines are based on theoretical assumptions, our aim was to establish the safety and tolerance of shortening the recommended interval between vaccinations. During two influenza seasons, 44 adult patients, with a history of systemic allergic reactions after a Hymenoptera sting, underwent 58 simultaneous allergen and seasonal influenza vaccinations (study group) while in the maintenance phase of venom immunotherapy (VIT). The control group consisted of 57 healthy adults who were vaccinated against influenza only. The conditions of the patients were monitored during hospital visits, and via telecommunication methods to evaluate the safety and tolerance of the procedure. Within the study group, there were no immediate or delayed allergic reactions after vaccinations. The presence of common, adverse influenza vaccine reactions among study group patients (29%) and control group patients (32%) did not differ significantly (*p* = 0.841). We did not observe a difference in the frequency of various adverse reactions in either group or a dependence of previous vaccinations against influenza on the occurrence of adverse reactions. The most frequent occurrences were local adverse reactions. All adverse reactions were resolved without treatment. These findings demonstrate the safety and tolerance of an influenza vaccination and Hymenoptera venom immunotherapy administration during one visit.

## 1. Introduction

The prevalence of allergic diseases is on the rise in both children and adults, affecting nearly one third of the global population [1]. The only causal therapy for allergic patients is allergen immunotherapy (AIT). This method relies on injections, in increasing doses, of an allergen extract in order to induce tolerance to the allergen [2]. Allergen immunotherapy is a time-tested, evidence-based, safe treatment for respiratory and venom allergies that is recommended by many guidelines [3]. However, the benefits of immunotherapy must be considered against the actual risks of local, systemic, or even rare life-threatening systemic allergic reactions [4]. The rate of immunotherapy-related systemic reactions of varying severity is relatively low (0.1–0.2%) [5]. Most systemic reactions occur within 30 min following the injection of an allergen extract [3]. To lower the risk of the occurrence of side effects, immunotherapy guidelines recommend a thorough pre-injection screening, as well as withholding injections from patients with active comorbidities, uncontrolled asthma, and infections. It is also advised to repeat the previous dose, or modify it, for individual patients who are assessed to be at greater risk of a systemic reaction [3,4]. A possible negative influence of beta and angiotensin-converting-enzyme (ACE) blockers on the safety of allergen immunotherapy has been considered lately. Recent clinical studies, however, have liberalized the use of cardiac medications in insect venom immunotherapy (VIT).

Another safety aspect regarding VIT, which is still not verified in well-designed studies, is the concurrent administration of allergen vaccines and vaccines against infectious agents, such as the annual influenza vaccine. Both types of vaccines affect the immunological system, but their purposes and mechanisms are different. Because of a lack of randomized, prospective studies on the concurrent administration of these two types of vaccines, and due to the concern that a “double stimulation” of the immune system may impair the tolerance of both vaccines and increase the occurrence of concomitant side effects, some guidelines suggest separating both injections. For example, the European Academy of Allergy and Clinical Immunology (EAACI) advocates separating both injections by at least one week. While this recommendation is based on theoretical assumptions, single retrospective clinical studies suggest that the administration of allergen and protective vaccines on the same day is safe [6]. Concomitant vaccinations are more comfortable for patients and make it possible to administer vaccines at a proper time. This issue concerns children, especially, but adults as well, e.g., those who need to get an annual influenza vaccine.

Influenza is a common viral infectious disease which can be prevented by applying vaccines. The effectiveness of vaccines may vary, as it depends, for example, on matching the vaccine with the circulating influenza viruses, as well as the patient-specific factors. According to many vaccination programs, there is a need for annual vaccinations against influenza in patients with an increased risk of severe influenza and its complications, as well as in all patients over 6 months of age because of epidemiological reasons. The significance of an annual influenza vaccination is supported by many evidence-based studies. It is confirmed that the influenza vaccine reduces the risk of influenza illness, influenza-associated hospitalizations among children as well as adults, and the risk of influenza-related death in children [7,8,9]. It is also known that the influenza vaccine can protect unvaccinated people (cocoon strategy) [10]. As there is a need to repeat the vaccination every year, it can be problematic for patients undergoing venom immunotherapy to schedule a vaccination appointment at the right time.

Influenza vaccinations appear to be particularly important during the Coronavirus Disease 2019 (COVID-19) pandemic because, by reducing the number of influenza-infected patients and the influenza mortality rate, the capacity of health systems can be preserved.

The aim of our study is to assess if the administration of allergen and protective vaccines during one visit is a safe option, and, additionally, if the reduction in time between the two injections to 30 min impairs the tolerance of both vaccines. Based on the fact that allergen vaccines and vaccines against infectious diseases have different mechanisms of action, we assumed that the suggested simultaneous procedure is safe for patients. We decided to check the incidence of adverse effects to evaluate the safety of double vaccination.

Additionally, changes in the concentration of IgM and IgG antibodies against influenza A and B viruses, at the end of the influenza season compared to the pre-vaccination or the beginning of the influenza season, were examined.

## 2. Materials and Methods

### 2.1. Database and Cohort

#### 2.1.1. Study Group

This is a prospective study conducted from 2017 to 2019 on a group of adult patients (*n*= 58) treated in the Department of Internal Medicine, Pneumology, and Allergology in Wrocław due to systemic symptoms of a Hymenoptera venom allergy (HVA-SYS). There were 58 double vaccine interventions involving 44 patients, and 14 out of 44 patients took part in the research during both influenza seasons (2017/2018 and 2018/2019). The study included patients with a history of systemic allergic reaction after a Hymenoptera sting (I–IV grade according to the Mueller classification), who are in the maintenance course of venom immunotherapy (VIT). All patients received the same dose (100 mcg) of venom vaccine (Alutard SQ, Alk Abello, Denmark). To evaluate the serologic evidence of contact with influenza virus, A and B serum samples were obtained from study group patients before administering the influenza vaccine, 4–8 weeks after the procedure (during the next visit resulting from the course of immunotherapy), and at the end of the influenza season (Figure 1). Serum samples were examined by Euroimmun ELISA (Enzyme-Linked Immunosorbent Assay) (Lubeck, Germany) Test Systems—IgM Antibodies against Influenza A (EI 2691-9601 M), Influenza B (EI 2692-9601 M), and IgG Antibodies against Influenza A (EI 2691-9501 G), Influenza B (EI 2692-9501 G) and by hemagglutination inhibition test-influenza A(H1N1).

The inclusion criteria to the study were:
a previous course of venom immunotherapy without any complications, and a stable maintenance dose of venom;a lack of contraindications to administering a subsequent dose of venom extract;the wish to be vaccinated against influenza.

The exclusion criteria were:
pregnancy or breastfeeding;mastocytosis;severe and uncontrolled asthma;any medical contraindications to both vaccinations.

#### 2.1.2. Control Group

The control group consisted of healthy patients (*n* = 57) vaccinated by the authors, within another study, only against influenza during influenza seasons 2016/2017, 2017/2018, and 2018/2019.

Detailed data on study group patients and control group patients is summarized in Table 1 and Table 2.

All the participants were informed about the possible side effects of both vaccines and signed a written informed consent document. All the patients were enrolled by trained medical personnel.

The study was approved by the Ethics Committee Medical University of Wrocław (731/2017).

### 2.2. Study Design

During scheduled visits at the clinic, patients were informed about the study and the possibility to get vaccinated against influenza. They were examined by medical personnel and were obligated to sign a written consent. Every day, 8–12 patients had an appointment in the clinic. We offered participation in our study to approximately 400 patients, but only 10–20% of patients admitted every day agreed to take part in the research.

Each patient from the study group received a proper venom vaccine (wasp or bee venom Alutard SQ, ALK Abello, Denmark) and influenza vaccine (Vaxigrip Tetra, Sanofi Pasteur, adequate for each influenza season) during one visit at the hospital and after a 30 min interval.

The injections were administered in opposite arms. Before the injections, patients were examined by medical personnel to exclude any contraindications for venom immunotherapy or the influenza vaccine. After the second injection, the patients were observed for 30 min in case any side effects appeared. Before leaving the hospital, patients were asked to immediately inform the personnel in case of any adverse reactions. Patients were monitored via telephone communication for 7 days after the injections.

Each patient from the control group only received the influenza vaccine (Vaxigrip for season 2016/2017 and Vaxigrip Tetra adequate for seasons 2017/2018 and 2018/2019). Participants were monitored for 7 days via telecommunication systems for the appearance of any adverse reactions.

All the patients were interviewed according to a standardized questionnaire.

The evaluation of the safety of the allergen and influenza vaccinations was assessed by the number and frequency of local and systemic adverse reactions in both the study and control groups.

The adverse reactions were categorized as immediate and delayed. An immediate reaction was defined as occurring during the first 30 min after the vaccine administration, and a delayed reaction was when the onset occurred within 30 min and 7 days after the intervention.

Local reactions were defined as pain, redness, swelling, bruising, and/or hardness of the injection site and were assessed by measuring the diameter of the skin lesion.

Systemic allergic adverse reactions were defined and classified according to the grading system recommended by EAACI: 0—no symptoms; 1—unspecific symptoms such as headache, arthralgia, and discomfort; 2—mild rhinitis/asthma responding well to pharmacological treatment; 3—non-life-threatening systemic reactions (urticaria, angioedema, or severe asthma, responding well to treatment); 4—anaphylactic shock [11].

We also monitored adverse reactions that are common for influenza vaccine [12,13,14] if they occurred within 7 days after injections: fever (body temperature above 38.5 °C), fatigue, malaise, muscle pain, joint pain, and influenza-like symptoms (defined as two or more of mentioned above).

Statistical significance among groups was calculated by the Fisher exact test. Values of *p* < 0.05 were considered significant.

Serologic evidence of exposure to influenza virus A and B, or the vaccination, was measured in the study group by seroconversion rate, and was assessed for strain A and B. The seroconversion rate was defined as the percentage of participants with negative/ border values of ELISA IgM titer during the first visit and positive values of ELISA IgM titer during the second (4, 6, or 8 weeks after vaccination) or third visit (at the end of each influenza season), or by doubling ELISA IgG titer after vaccination. During season 2017/2018, we compared these findings with the serological status of 25 patients undergoing VIT who were not vaccinated against influenza (two serum samples were obtained and tested at the beginning and at the end of influenza season).

## 3. Results

### 3.1. Safety

We observed neither immediate local nor systemic allergic reactions after the administration of insect venom vaccine in the study group. Within the first 30 min after the administration of the influenza vaccine, no other adverse reactions occurred in the group vaccinated with both allergen and influenza. No delayed local reactions were reported by patients during the first 24 h after both injections.

The frequency and severity of any observed adverse reactions did not differ between the group vaccinated with allergen and influenza and the group vaccinated only with influenza (Table 3). In the time of 7 days of observation, 17 patients (29%; 17/58) vaccinated with both vaccines reported adverse reactions that are common for influenza vaccination, and the majority of them (59%; 10/17) were vaccinated against influenza for the first time. The occurrence of adverse reactions that are common for influenza vaccination did not differ between patients vaccinated against influenza for the first time and patients (23%; 7/31) vaccinated against influenza more than once (*p* = 0.260). Among 14 patients, who underwent double intervention in both seasons, only 2 of them (14%; 2/14) reported adverse reactions during the second season of influenza vaccination.

There were 18 patients <32%> from the control group who reported adverse reactions common for influenza vaccine. Among control group patients, we observed adverse reactions in 30% (8/27) of patients vaccinated against influenza for the first time and in 33% (10/30) of patients vaccinated against influenza more than once. As in the study group, the frequency and severity of these reactions did not differ between patients vaccinated for the first and many times (*p* = 0.781).

In general, the number of patients who reported adverse reactions did not differ among the study group (29%; 17/58) and the control group (32%, 18/57), *p* = 0.841. All adverse reactions that appeared on the second day after intervention were mild, transient (lasting 2–3 days), and resolved without any treatment. Adverse reactions were reported during scheduled telephone conversations, none of the patients contacted the researchers personally.

### 3.2. Serologic Evidence of Exposure to Influenza Virus A and B Antigens in a Single Season

In Table 4, we summarize the number of patients from the study group with anti-influenza antibody seroconversion during the seasons 2017/2018 and 2018/2019.

We compared the 2017/2018 season findings with the influenza serological status of 25 patients undergoing VIT who were not vaccinated against influenza (Table 3). We observed influenza A seroconversion in 12/25 = 48% patients (*p* = 0.185) and influenza B in 8/25 = 32% patients (*p* = 0.067).

At the beginning of the 2017/2018 season, all the patients (33/33) among the study group had positive results of IgG against influenza A, and 32/33 patients had positive results of IgG against influenza B. In the group of 25 patients undergoing VIT who were not vaccinated against influenza, all the patients were positive for influenza A and B IgG at the beginning of influenza season.

### 3.3. Influenza Hemagglutination Inhibition Test

All serum samples were positive for hemagglutination inhibition.

## 4. Discussion

### 4.1. Summary of the Results

In this study, we showed that the administration of the influenza vaccine 30 min after the Hymenoptera insect venom allergen vaccine was safe. Such a means of vaccination was well tolerated by the patients, and it did not increase the percentage of both early and late side effects. Furthermore, double vaccination did not increase the risk of adverse reactions during the second year of the intervention.

### 4.2. Clinical Importance of the Results

We believe that our findings have important implications for everyday clinical practice. The administration of both vaccine types in one day, during one visit to a doctor’s office, might significantly increase the number of patients vaccinated against influenza. This is particularly significant for a country like Poland, where vaccination against influenza is at a very low level, especially among adults. This pattern of vaccination is also highly recommended in the SARS-CoV-2 (severe acute respiratory system coronavirus 2) pandemic because it reduces the number of visits in health care system units.

### 4.3. Current Guidelines Regarding Vaccinations in Patients Undergoing Allergen Immunotherapy

The results of this study are unique, as we focused on the safety aspects of vaccination during allergen immunotherapy. The current subcutaneous immunotherapy (SCIT) guidelines in this topic are based mainly on theoretical assumptions and expert opinions [4]. As far as we know, there have not been any publications describing prospective studies on the safety of SCIT and other vaccinations.

According to the general recommendations of the Advisory Committee on Immunization Practices (ACIP), allergen immunotherapy is not a contraindication to administration of vaccines preventing infectious diseases [15]. Existing guidelines recommend 7 day intervals between the administration of these two types of vaccines. These guidelines were established upon experts’ opinions, resulting from concerns about the risk of multiplying side effects, especially systemic reactions, which may occur after the administration of each vaccine [4]. Manufacturers of some allergen vaccines dedicated to subcutaneous administration, in a summary of product characteristics (SmPC), advise maintaining the intervals between the two types of vaccines [16,17,18,19], and, sometimes, it is even necessary to reduce the dose of an allergen vaccine (for example, after an anti-tetanus vaccination). No such recommendations concern vaccines used in sublingual immunotherapy (for example Grazax) [20].

### 4.4. Previous Data Regarding Safety of Simultaneous Vaccinations

Our results, regarding the safety of this procedure, correlate fairly well with the retrospective study done by Ullrich et al., which also supports the safety of using two different types of vaccines (SCIT and infectious vaccine against influenza, pneumococcal bacteria, and viral hepatitis B) administered in one day. An analysis of the data gathered from 95 patients, included in this study, showed no significant rise in the side effects using this kind of vaccination protocol [6]. In a study performed by Garner-Spitzer et al., concerning immune response to vaccinations in allergic patients undergoing allergen-specific immunotherapy, it was confirmed that protection against the TBE (Tick-borne encephalitis) virus was not impaired in patients receiving SCIT compared to allergic patients with symptomatic or no treatment and healthy controls [21].

### 4.5. Tolerance of Combination of Allergen and Influenza Vaccines

The other intriguing aspect of simultaneous vaccination is its impact on immunological reactions. There used to be a lot of concern whether vaccination protocols including multiple vaccines administration would affect the immunity of each vaccine. In the 1940s, the FDA (Food and Drug Administration) introduced, to wide clinical use, one of the first combined vaccines, consisting of three pathogens: diphtheria/tetanus/pertussis (DTP) [22]. Since then, the administration of combined vaccines has become commonly accepted and applied. The administration of the Hymenoptera venom vaccine did not affect the tolerance of the anti-influenza vaccine. It is because the immune system response mechanisms in allergen vaccines are specific to the allergen and limited to the allergen-specific clones of lymphocytes, which reduces the risk of possible interference with immune response to the anti-influenza vaccine. We observed that the prevalence, and the type, of post-vaccination reactions presented after the administration of the anti-influenza vaccine in study group patients did not differ significantly from those observed in the control group.

### 4.6. Safety Measures Implemented in our Study

We would like to note that our observations, regarding the safety and tolerance of vaccination protocol and applying 30 min intervals between each vaccine administration, are valid to patients receiving maintenance doses in the course of VIT immunotherapy and to those who have no history of side effects after previous injections. According to existing studies, in the group of the patients selected this way, the estimated risk of systemic allergic reaction is 0.26% per injection, which is 10 times lower than in the initial phase of VIT (2% per injection) [23].

Our additional safety measure was applying subsequent injections in 30 min pauses because this is the time when serious allergic reaction is usually present. Severe influenza vaccine adverse events also usually occur within 30 min post-injection.

Over the last few years, there has been a breakthrough in vaccinating patients with an egg allergy against influenza. According to current knowledge, the influenza vaccine is safe even in egg allergic patients, and there is no need to use any other non-standard precautions during vaccination of such patients [24].

### 4.7. Differences of Influenza Serological Status among Patients from Study and Control Group

Our results regarding the serological status of people undergoing VIT indicate serologic responses to influenza vaccination and contact with wild-type influenza virus strains during the whole influenza season. We did not observe statistically significant differences between the study group and comparison group regarding influenza A and B seroconversion rate. Almost all patients in both groups were positive for influenza A and B IgG before the influenza season. These findings may indicate that ELISA tests do not represent a correct method of assessing the antibody response of the influenza vaccine.


Positive results of hemagglutination inhibition tests indicate that all patients had prior contact with the influenza virus.


This is a first prospective study confirming the safety of influenza and Hymenoptera venom allergen vaccination during one visit. However, we are aware that our research is limited, as the experiment was not blinded.

## 5. Conclusions

Our data indicates the safety and tolerance of an influenza vaccination and Hymenoptera venom immunotherapy administration, during one visit with 30 min intervals, in patients undergoing the maintenance phase of immunotherapy. We demonstrated that simultaneous vaccination with an allergen and influenza vaccination does not increase the risk of local and systemic adverse reactions, and it is well tolerated by the patients.

This procedure was safe and well-tolerated when repeated the next year. These findings could significantly simplify the procedure of annual influenza vaccinations in allergic patients. Further research regarding the immunogenicity of the influenza vaccine, as well as the safety and tolerance of other vaccines against infectious diseases applied by adult patients (vaccines against tick-borne encephalitis, pneumococcal or meningococcal infections) in allergic patients treated with immunotherapy is needed, but our results are promising.

## Figures and Tables

**Figure 1 vaccines-09-00344-f001:**
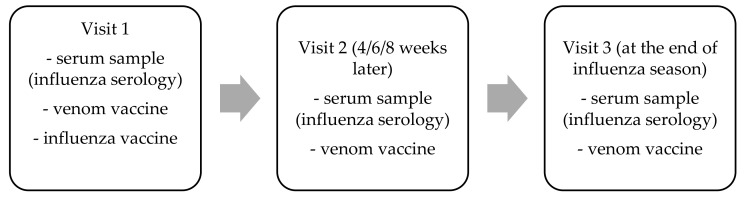
Study scheme among study group patients.

**Table 1 vaccines-09-00344-t001:** Demographic data of the study group patients, total *n* = 58 (influenza seasons 2017/2018 and 2018/2019), and control group patients, total *n* = 57 (influenza seasons 2016/2017, 2017/2018, and 2018/2019).

	Study Group Patients (*n* = 58)	Control Group Patients (*n* = 57)
Sex		
Female	22 (37.93%)	22 (38.60%)
Male	36 (62.07%)	35 (61.40%)
Age (years)		
Range	19–80	22–70
Median	47.5 (Q1 = 35, Q3 = 58)	24 (Q1 = 23, Q3 = 26)
Mean	46.4 (SD = 14.8)	30 (SD = 14.6%)
History of previous vaccinations against influenza		
None	27 (46.55%)	27 (47.37%)
One	16 (27.59%)	6 (10.53%)
Two and more	15 (25.86%)	24 (42.10%)

**Table 2 vaccines-09-00344-t002:** Characteristics of study group patients (*n* = 58).

Allergy to: (*n* = 58 Patients)	
Bee venom	12
Wasp venom	45
Bee and wasp venom	1
**Severity grade of HVA (*n* = 58 patients)**	
1	1
2	8
3	22
4	27
**Duration of the maintenance phase of VIT (years)**	
Range	0–5.5
Median	1.75
Mean	2

**Table 3 vaccines-09-00344-t003:** The number of patients among the study and control groups reporting adverse reactions after vaccination.

	Study Group (*n* = 58)	Control Group (*n* = 57)	*p*-Value
Allergic reactions	0	0	-
Administration site conditions	10	7	0.601
General disorders:			
-fatigue	2	4	0.436
-malaise	0	2	0.244
-fever	0	1	0.496
-muscle pain	0	1	0.496
-headache	3	1	0.618
-influenza-like symptoms	3	4	0.717

**Table 4 vaccines-09-00344-t004:** Seroconversion rate of the influenza vaccine among study group patients during both influenza seasons.

	Season 2017/2018 (*n* = 33 Patients)	Season 2018/2019 (*n*= 25 Patients)
influenza A	22/33 = 67%	10/25 = 40%
influenza B	19/33 = 58%	12/25 = 48%

## Data Availability

The data presented in this study are available within the article.

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
