# Peer review of "Simultaneous Influenza Vaccination and Hymenoptera Venom Immunotherapy Is Safe"

_vaccines, 2021, doi:10.3390/vaccines9040344_

Round 1

Reviewer 1 Report

The author described a quite good method to demonstrate safety and tolerance of influenza vaccination and Hymenoptera venom immunotherapy administration during one visit. Allergen immunotherapy (AIT) is a well-recognized treatment for allergic disease that builds immune tolerance through the administration of specific allergens. It is the only known therapy to alter the natural history of respiratory allergy by targeting allergen-activated immune cells such as T helper 2 cells (Th2) and regulatory T cells (Treg).

We suggest the author would test the challenges of using peripheral eosinophilia alone as a biomarker.

If could test more various cytokines would be better.

As the author mentioned, we worried about that the research is limited as the experiment was not blinded.

And we also suggest the author reorganize all the table in the manuscript.

Author Response

Dear Reviewer,

thank you for comments on our manuscript entitled "Simultaneous influenza vaccination and Hymenoptera venom immunotherapy is safe".  

We appreciate suggested modifications. We tried to address all of them.

Below we provide the point-by-point responses. All modifications in the manuscript have been highlighted in yellow.

General comment: The author described a quite good method to demonstrate safety and tolerance of influenza vaccination and Hymenoptera venom immunotherapy administration during one visit. Allergen immunotherapy (AIT) is a well-recognized treatment for allergic disease that builds immune tolerance through the administration of specific allergens. It is the only known therapy to alter the natural history of respiratory allergy by targeting allergen-activated immune cells such as T helper 2 cells (Th2) and regulatory T cells (Treg).

Response: Thank you for your opinion.

Comment 1: We suggest the author would test the challenges of using peripheral eosinophilia alone as a biomarker.
If could test more various cytokines would be better.

Response: Thank you very much for your suggestions. The primary objective of our study was to assess the safety of administering concurrently allergen vaccine and influenza vaccine. Occurrence of adverse events was assessed based on clinical symptoms during telephone conversations. Our observations confirm the hypothesis that allergen vaccines and anti-flu vaccines can be safely administered with a 30 minutes’ interval.

Our study did not undertake assessments of peripheral eosinophilia or cytokine levels since we are convinced that a single exposure to Hymenoptera venom allergen combined with a flu vaccine shot can affect serological and/or cellular response to flu vaccine but has no impact on venom allergen tolerance in patients in the maintenance phase of venom immunotherapy (VIT). This phase of VIT is characterized by induction of regulatory T cells (Treg) that in turn perform pluripotent actions via multiple suppressor factors including IL10, TGFb,IL35, IL10R, CTLA4, TGFbR, PD1, and HR2, affecting the pro-allergenic milieu. It is also the phase of VIT in which suppression of the effector cells of allergic inflammation — mast cells, basophils, and eosinophils – is observed. We assumed that the said mechanisms are not interfered with by a flu vaccine administration.

Comment 2: As the author mentioned, we worried about that the research is limited as the experiment was not blinded.

Response: Thank you for your comment. Immunotherapy could not be blinded as during our research the patients were undergoing venom immunotherapy for a long time and they knew the whole procedure. Preparation of influenza vaccine placebo was beyond authors’ possibility. The advantage of taking part in our experiment included free influenza vaccine administered during scheduled visit in Allergology Clinic. Despite such encouragement 80-90% of patients admitted every day didn’t agree to take part in our research. Till now nobody published results of prospective, controlled research concerning simultaneous administration of allergen vaccine and influenza vaccine. Our experiment was not blinded as it is typical for initial phase of clinical trials. What is more, symptoms of allergic reactions due to subcutaneous immunotherapy and vaccinations are clear and hard to induce spontaneously, so we are sure that there is no risk of falsifying the results even if the experiment was not blinded.

Comment 3: And we also suggest the author reorganize all the table in the manuscript.

Response: Thank you for your comment. We reorganized the tables in the manuscript to make them more readable and understandable:

- We divided data in Table 1 into two smaller tables (Table 1a and Table 1b).

- We changed the statement introducing Table 1: Detailed data on study group patients and control group patients is summarized in Table 1. The revised text reads as follows on lines 128-129: Detailed data on study group patients and control group patients is summarized in Table 1a and Table 1b.

- We changed a description of Table 1, adding the name (Table 1a).

- We added a description of Table 1b: Table 1b. Characteristics of study group patients (N=58).

- In Table 2. we made the text in first column left aligned.

- We changed the description of Table 2: Adverse reactions among study and control group patients. The revised text reads as follows on lines 232-233: The number of patients among study and control group reporting adverse reactions after vaccination. 

We did not make any changes of organization of Table 3.

We thank you for your precious time spent on reviewing our paper and providing valuable comments. We hope the manuscript after careful revisions will meet your high standards. The authors welcome further comments.

Yours sincerely,

Authors

Reviewer 2 Report

I read with interest the manuscript submitted to me for review. There is no doubt that any initiative to implement vaccinations is welcome. I have doubts about the real novelty of the results achieved in the study.

Major

  1. In the results section the reference to the tables should appear in the text.
  2. What do lines 224-227 refer to? If they refer to Table 3 there is some error.
  3. The reference to seropositivity towards flu A and B is important news if it was a study on the antibody response after vaccination and during the flu season, but in this context it has no meaning.
  4. I would suggest shortening the discussion, the length of which is excessive in relation to the results.

Minor

  1. The references, both in the text and in the list, are not according to the editorial instruction.
  2. The text is in English, so the comma to define a fraction should be replaced by a period.

Author Response

Dear Reviewer,

thank you for comments on our manuscript entitled "Simultaneous influenza vaccination and Hymenoptera venom immunotherapy is safe".  

We appreciate suggested modifications. We tried to address all of them.

Below we provide the point-by-point responses. All modifications in the manuscript have been highlighted in yellow.

General comment: I read with interest the manuscript submitted to me for review. There is no doubt that any initiative to implement vaccinations is welcome. I have doubts about the real novelty of the results achieved in the study.

Response: Thank you for your opinion. Our research is a first prospective study concerning safety of simultaneous vaccinations in patients undergoing allergen immunotherapy. There are no similar prospective studies concerning this topic at all (17.03.2021).  If there were any studies confirming our thesis, then the recommendations of scientific societies would be based on such publications and not opinions. This long suggested interval between allergen immunotherapy and other vaccinations is probably the cause of preservative attitude towards vaccinations, what explains high ratio of refusals to take part in our research.

Major comment 1: In the results section the reference to the tables should appear in the text.

Response: Thank you for your comment. In the revised manuscript, we have modified the results section adding refernece to the tables:

- we added a sentence: In Table 3 we summarized the number of patients from study group with anti-influenza antibodies seroconversion during seasons 2017/2018 and 2018/2019. (line 241 and 242)

- we modified the description of Table 3 to make it clearer, the new description is as follows:  Seroconversion rate of influenza vaccine among study group patients during both influenza seasons. (lines 237 and 238)

Major comment 2: What do lines 224-227 refer to? If they refer to Table 3 there is some error.

Response: Thank you for pointing this out, this paragraph may be confusing. Lines 224-227  do not refer to Table 3, it is new data concerning another group of patients. Table 3 refers to patients from study group and their seroconversion rate. Lines 224-227 refer to 25 patients undergoing allergen immunotherapy who were not vaccinated against influenza, they were monitored to assess their seroconversion rate. We wanted to see if there are any differences in seroconversion rate among  patients undergoing allergen immunotherapy who were vaccinated and were not vaccinated against influenza.

Major comment 3: The reference to seropositivity towards flu A and B is important news if it was a study on the antibody response after vaccination and during the flu season, but in this context it has no meaning.

Response: Thank you for your opinion, we agree that influenza A and B seropositivity is secondary topic of our study. We planned serological tests to assess if allergen immunotherapy would have any impact on influenza vaccine humoral response. As it turned out, majority of our patients had antibodies from previous seasons and the seroconversion rate after vaccination was similar in both groups. We demonstrated that exposure to Hymenoptera venom did not change the humoral response of influenza vaccination. We can add this conclusion to our discussion section. On the other hand, as the results of serological tests are not connected with safety of the whole procedure we agree to remove this part of our article.   

Major comment 4: I would suggest shortening the discussion, the length of which is excessive in relation to the results.

Response: Thank you for your opinion. The discussion may look very long, but we wanted to relate to many aspects of safety of allergen immunotherapy and influenza vaccination (allergic reactions, other adverse effects, vaccinating patients with egg allergy), current recommendations in this topic, other studies concerning this problem and benefits of simultaneous vaccination in everyday practice. We hope that this discussion will help to understand sense and importance of our research.  As there is a new approach to vaccinating people with egg allergy [Greenhawt, M.; Turner, P. J.; Kelso, J. M. Administration of Influenza Vaccines to Egg Allergic Recipients: A Practice Parameter Update 2017. Ann. Allergy, Asthma Immunol. 2018, 120(1), 49–52] and there is no immunotherapy against allergy we agree to remove this paragraph. 

Minor comment 1: The references, both in the text and in the list, are not according to the editorial instruction.

Response: Thank you for your reminder, the references are now changed according to the editorial instruction.

Minor comment 2: The text is in English, so the comma to define a fraction should be replaced by a period.

Response: Thank you for this reminder, we have replaced all wrongly used commas accordingly.

We thank you for your precious time spent on reviewing our paper and providing valuable comments. We hope the manuscript after careful revisions will meet your high standards. The authors welcome further comments.

Yours sincerely,

Authors

Reviewer 3 Report

The manuscript entitled “Simultaneous influenza vaccination and Hymenoptera venom immunotherapy is safe.” This work is merit for publication at Vaccines after some major modification. So I have some points that may help to improve the work as follows:

1-Abstract is good but need more explain about the main aim of work

2- The introduction should be extended to discuss the hypothesis and research questions in details. Additionally, the introduction should cover the recent literature related to this subject.

3- Material and methods

The methodologies should be explained in details so that the results are reproducible.

4-Results

The results are clear and important.

5-Discussion
The discussion section still needs improvement, and should be linked to the findings of the previous reports on this topic.

6- The conclusion

A section for conclusions need more explain and should include the most significant findings and future works only.

7- English writing should be checked by a native English speaking expert.

Author Response

Dear Reviewer,

thank you for comments on our manuscript entitled "Simultaneous influenza vaccination and Hymenoptera venom immunotherapy is safe".  

We appreciate suggested modifications. We tried to address all of them.

Below we provide the point-by-point responses. All modifications in the manuscript have been highlighted in yellow.

General comment: The manuscript entitled “Simultaneous influenza vaccination and Hymenoptera venom immunotherapy is safe.” This work is merit for publication at Vaccines after some major modification. So I have some points that may help to improve the work as follows

Response: Thank you for your opinion. We have gone through your comments carefully and tried to address all of them. We hope the manuscript has been improved accordingly.

Major comment 1: Abstract is good but need more explain about the main aim of work.

Response: Thank you for your comment. In our abstract we wrote that the purpose of our work is to determine safety and tolerance of simultaneous vaccination against Hymenopera venom and influenza. To underline the aim of our study we added one more sentence:  As current guidelines are based on theoretical assumptions, our aim was to establish safety and tolerance of shortening recommended interval between vaccinations. (lines 13-15)

Major comment 2: The introduction should be extended to discuss the hypothesis and research questions in details. Additionally, the introduction should cover the recent literature related to this subject.

Response: Thank you for your comment, we extended the introduction as you suggested, adding new sentences concerning our hypothesis: Basing on the knowledge that allergen vaccines and vaccines against infectious diseases have different mechanisms of action we assumed that suggested simultaneous procedure is safe for the patients. We decided to check the incidence of adverse effects to evaluate safety of double vaccination. (lines  83-87)

We didn’t find any other than mentioned (line 298, reference [6]) examples of studies concerning safety of vaccinations among patients undergoing allergen immunotherapy in previous literature.

Major comment 3: The methodologies should be explained in details so that the results are reproducible.

Response: Thank you for your proper remark, we have corrected this problem.

- Specific information regarding enrolment of the patients to our study could be described as follows: “During scheduled visits in the clinic, patients were informed about the study and the possibility to get vaccinated against influenza. They were examined by medical personnel and were obliged to sign a written consent. Every day 8-12 patients had an appointment in the clinic. We offered participation in our study approximately 400 patients, only 10-20% of patients admitted every day agreed to take part in the research.” We inserted this paragraph to main text (lines 146-150)

- Should we include specific information regarding execution of laboratory tests?

Major comment 4: The results are clear and important.

Response: Thank you for your opinion.

Major comment 5: The discussion section still needs improvement, and should be linked to the findings of the previous reports on this topic.

Response: Thank you for your opinion. To improve our discussion section and make it  more structured we added a number of subsections with informative headings, new topics are as follows:

- 4.1. Summary of the results (line 256)

- 4.2. Clinical importance of the results (line 263)

- 4.3. Current guidelines regarding vaccinations in patients undergoing allergen immunotherapy (lines 272-273)

- 4.4. Previous data regarding safety of simultaneous vaccinations (line 292)

- 4.5. Tolerance of combination of allergen and influenza vaccines (line 304)

- 4.6. Safety measures implemented in our study (line 319)

- 4.7. Differences of influenza serological status among patients from study and control group (lines 355-356)

 We believe that now the argument is clearer, but we would welcome comments on particular sections and headings if you have further concerns.

Our results are linked to the previous reports on this topic (section 4.4. Previous data regarding safety of simultaneous vaccinations).

Major comment 6: A section for conclusions need more explain and should include the most significant findings and future works only.

Response: Thank you for your opinion. We have modified the conclusions section focusing on the most significant findings and future works adding new passages:

- We demonstrated that simultaneous vaccination with allergen and influenza vaccination doesn't increase the risk of local and systemic adverse reactions and is well tolerated by the patients. (lines 353-355)

- (...) as well as safety and tolerance of other vaccines against infectious diseases applied by adult patients (vaccines against Tick-borne encephalitis, pneumococcal or meningococcal infections) (lines 358-361)

Major comment 7: English writing should be checked by a native English speaking expert.

Response: Thank you for your comment, our manuscript have been checked by a Canadian trained in biological studies. We have carefully revised our work once again to eliminate any grammar, spelling and style errors.

We thank you for your precious time spent on reviewing our paper and providing valuable comments. We hope the manuscript after careful revisions will meet your high standards. The authors welcome further comments.

Yours sincerely,

Authors

Round 2

Reviewer 2 Report

I have not further comments. The authors have answered comprehensively to my suggestions.

Reviewer 3 Report

The authors improved the overall quality of their manuscript, I think that now it can be accepted for publication on Vaccines as it is.